# Perception of School Violence: Indicators of Normalization in Mapuche and Non-Mapuche Students

**DOI:** 10.3390/ijerph20010024

**Published:** 2022-12-20

**Authors:** Flavio Muñoz-Troncoso, Isabel Cuadrado-Gordillo, Enrique Riquelme-Mella, Edgardo Miranda-Zapata, Eliana Ortiz-Velosa

**Affiliations:** 1Faculty of Education, Universidad Católica de Temuco, Temuco 4810296, Chile; 2Faculty of Education and Psychology, Department of Psychology and Anthropology, Universidad de Extremadura, 06071 Badajoz, Spain; 3Universidad Mayor, Temuco 4801043, Chile; 4Universidad Autónoma de Chile, Temuco 4810101, Chile; 5Faculty of Architecture, Art and Design, Universidad Católica de Temuco, Temuco 4810296, Chile

**Keywords:** school violence, school coexistence, intercultural education, Mapuche, questionnaire, emotions, factorial invariance

## Abstract

The current social and political scenario in Chile has opened up the debate on two centuries of usurpation and discrimination towards the Mapuche people. Educational centers are not oblivious to the social exclusion faced by indigenous children and young people, and this forms part of the phenomenon of school violence. This study explores the differences in perception between Mapuche and non-Mapuche students regarding school violence. The issue is the lack of knowledge regarding cultural variations in the perception of school violence in spaces of social and cultural diversity in the Mapuche context. This study describes the characteristics of school violence perceived by students in relation to differences based on ancestry and characterizes the variations in perception. A total of 1404 students participated from urban schools in the city of Temuco, Chile, aged 10 to 13 (M = 11.4; SD = 1.1) who completed the CENVI questionnaire. The confirmatory factor analysis (CFA) of the total sample and categories provides indexes that fit the proposed model. The omega coefficients provide internal reliability guarantees. This study tests configural, metric and scalar invariance for all the categories explored, and statistically significant differences are found between Mapuche and non-Mapuche students in the perception of physical and verbal violence, where the Mapuche student perceives more violence. Results are discussed based on existing research on education in spaces of social and cultural diversity in the Mapuche context, with research into elements that can help explain the findings.

## 1. Introduction

Violence has been observed for decades as a historical construct associated with domination [1,2,3,4,5]. For some theorists, this goes beyond individual actions of an instinctive or irrational nature [6]. Undoubtedly, this leads to the delegitimization of the other that is similar, giving rise to submission and vulnerability in contexts that frame violence as a socially acceptable phenomenon [7]. This is due to the naturalization of violent behaviors in the daily life of people as social beings [8].

From a sociological approach presented several decades ago and still maintained by Galtung [9], violence emerges as: (1) direct, when manifested physically or verbally in a tangible way; (2) structural, if it is part of the political and economic system of a society; and (3) cultural, when exercised through religion, laws, language and other elements of culture, which legitimize direct, structural violence. The author alludes to a triangle that is formed by these types of violence, which can be triggered by any of them and can permeate the others. Since the violent structure is institutionalized and the violent culture is internalized, direct violence may be cyclical and ritual.

A social and historical analysis of violence carried out by Han [10] proposes that the causes of the phenomenon are external to the individual, but that they transit inward until they lodge inside the individual. The author maintains that today, the exercise of violence is considered illegitimate, to the degree that it is made invisible if it appears in society. The author proposes the concept of microphysical violence, composed of a triangle where its characteristics of internalization, automation and naturalization interact. This prevents a reflection on the exercise of violence with the consequent impossibility of avoiding its practice.

From a health perspective, the World Health Organization (WHO) [11] defines violence as the intentional use of physical force or power, threatened or actual, with the aim of causing physical or psychological harm. This organization considers that the origin of violence can be explained by multicausal factors such as family, community and cultural factors, as well as biological conditions and other individual elements that can affect its occurrence. Violence, then, is a multifactorial phenomenon that involves the aggressor, the victim and the social system [12]. It is important to highlight the psychological elements associated with experiences of violence, which imply that the victim can also be an aggressor. The case study by Navas and Cano [13] found that all victimized aggressors exercised more violence than non-victimized aggressors. Cuadrado and Fernández [14] refer to the aggressor-victim, who, upon seeing the diminished social, physical or psychological power that they initially used to cause harm, is exposed as a victim while also being an aggressor. The same aggressor-victim phenomenon occurs in the opposite sense. Cuadrado et al. [15] found some predictors of the co-occurrence of student victims of bullying who go on to become perpetrators of cyberbullying.

Violence is, indeed, a historical construct associated with domination [1,5], from a logic of the power of an individual or group over another individual or group [9]. The evolution of social and political systems has implied the emergence of violence in various forms and scenarios [10,15], such as the school setting.

According to Guajardo et al. [16], conceptually, school violence is built upon a shared ontological assumption, which considers the existence of a reality of school violence. It assumes the phenomenon as the composition of discrete levels, mobilized on a scale of inclusivity between categories, composed of relationships that are hierarchical in school but not in the broader setting. This vision assumes an ontological stratification composed in some cases by differentiated spaces and in others by autonomous spaces.

Guajardo et al. [16] found seven levels that affirm the ontological assumption of the ‘reality of school violence.’ The first four of these are: (1) individual, as an indivisible unit because it is the subject or person; (2) interpersonal relationships, since these imply interactions between individuals; (3) student groups; and (4) school. Until this point, the authors identify a hierarchy in the levels, wherein the school setting contains the other levels in what is considered to be the interior of the school. According to Guajardo et al. [16], the remaining levels, which do not demonstrate a hierarchy, are: (5) the school system; (6) educational policies; and (7) the context. These are considered external to the school itself. Figure 1 presents a graphic representation of this idea and shows the relationships that occur between levels.

Regarding the exercise of school violence itself, Cedeño [17] maintains that it is a form of interaction between students, which generally involves physical or psychological force. However, the study proposes that although in some cases the objective is to cause harm, there are other situations where the aggressors’ aim is to assert themselves within their environment. Research conducted by Cuadrado [18] revealed that some students perceive certain types of verbal aggression as a form of social interaction and, to a lesser extent, perceive it as associated with the intent to cause harm. Likewise, according to Yang et al. [19], for teenagers, cyberbullying is seen as a prank, in the context of forms of maladaptive humor. According to Cedeño [17], the use of violence is a learned form of conflict resolution, which means that it can potentially be eradicated through the learning of preferentially peaceful mechanisms. Menesini and Salmivall [20] argue that school violence as an illegitimate strategy for conflict resolution is a serious and complex phenomenon.

Research into school violence—and its effects—has focused on identifying the ways in which it manifests inside the school or in other spaces considered part of the school setting. Diverse research has identified the following types of violence: (1) verbal violence; (2) physical violence; (3) violence through social exclusion; (4) violence through technology; and (5) teacher violence towards students [4,21,22]. Although some of the mentioned manifestations of school violence may be more prevalent than others, the occurrence of any of these constitutes a risk to healthy school coexistence [5,7]. This is due to the negative psychosocial effects on students [23,24], potential harm to physical health [25] and the subsequent detrimental effect on learning [22,26,27].

Studies by López et al. [28] propose that social cognition does not explain the phenomenon of violence in its entirety. These authors argue that social cognition skills mediate the relationship between macrostructural variables (socioeconomic level) and microstructural variables (classroom climate) on school violence. They propose an integrated model with the factors and the organization of these for the study of school violence (see Figure 2).

The model seeks to understand violence among students in the school system as part of a complex scenario, where the factors that explain the phenomenon are part of the individual characteristics of victims and aggressors. These factors form part of a larger context, made up of other interrelated factors [28]. This ties individual actions with the interdependence of the context to which they belong. In this sense and in the context of this research, it is important to consider the individual and cultural factors of the subjects involved in experiences of violence in relation to their participation in spaces of interaction in the school system.

School violence is a reality, with hierarchical levels within the school environment and non-hierarchical levels in the external environment [16]. This is a non-peaceful form of conflict resolution, where aggressors in some cases aim to cause harm and in other cases to assert themselves in the environment [17]. Some students consider violence to be part of the social interaction with their classmates [18], implying a normalization of violence [15]. Diverse research refers to the concept of entrenched violence, whose effects can extend intergenerationally, thereby affecting future generations [29,30]. For this reason, violence is a phenomenon whose prevalence should be reduced, due to the harmful effects on the subjects who experience it [23,24] and on the academic results of the school system [22,26,27]. Different forms of manifestation or types of school violence can be identified, which are perceived by the students themselves [4]. Research on school violence must consider macrostructural and microstructural variables, which give rise to the interaction between factors and contexts [28].

## 2. Mapuche School Context

Schooling in the Mapuche context refers to knowledge transmission processes and the search for educational purposes in formal education, whose purpose has been the teaching of the world from the perspective of the dominant other. In relation to this, Muñoz and Quintriqueo [31] comment that the Mapuche people have experienced three types of schooling since the Hispanic invasion, specifically: (1) the evangelization (conquest era), understood as the teaching of the Christian faith to form the idea of submission to the King of Spain in obedience to God; (2) the missionary schools (colonial era), whose purpose was to train young Mapuche sons of *logko* (maximum political authority) as missionaries, to evangelize in their communities of origin; and (3) the public school (republican era), which fostered the formation of national identity to ensure the unity of the emerging State of Chile [32,33]. This colonialist education project sought to teach a minimum school curriculum focused on literacy, elementary mathematics and Spanish monolingualism. This project promoted the basic social inclusion of the Mapuche people into the working–peasant class of Chilean society, as subjects alienated from their own culture [34,35,36].

Based on the above, schooling in the Mapuche context has been a space of submission to the colonization of being, knowing, being able and doing in the rationality of the Mapuche people for their assimilation into Chilean society through a precarious inclusion model [37,38]. Porma [39] argues that this has been developed within the framework of colonial violence in historical Araucanía, which continues to the present day. This fact is reflected in a school that has neglected to develop the intellectual potential of Mapuche students for the conscious or unconscious purpose of keeping them in a lower social class of Chilean society [36,40]. The outcome of this is that in La Araucanía, scores on standardized tests implemented by the Education Quality Measurement System (SIMCE, Sistema de Medición de la Calidad de la Educación) are historically the lowest [41], while social vulnerability in elementary, secondary and higher education are among the highest, according to data from the Ministry of Social Development and Family [42]. These are all elements that can significantly affect the indigenous subject until adulthood, namely the prejudices addressed in the study by Saiz et al. [43], where meta-stereotypical labels and attributes—all negative—were found regarding how the Mapuche people perceive how they are seen by non-Mapuche people.

Among the difficulties involved in the implementation of state programs on intercultural education, the persisting baseline epistemic and social asymmetries between the state, indigenous peoples and non-indigenous society are ignored [44]. This makes implementation complex, considering the colonial and Eurocentric nature of the schooling institution, a fact that makes it difficult to intervene in educational projects, thus denying the possibility of intercultural teaching–learning processes [45]. Several studies have confirmed that in the Mapuche context, bilingual intercultural education (BIE) does not make the intended contribution, instead becoming an extension of the state’s monocultural and colonialist policy [46,47,48]. This is because it is a low-quality education that does not report improvements in learning and educational processes, it is only aimed at indigenous people, and there is a stigma that it hinders performance on the SIMCE. It is merely a functionalist interculturality to serve the purposes of the state [45,48,49]. Whalsh [50] and Bertely et al. [51] argue that the solution is to move towards a critical and decolonial interculturality that: (1) relativizes power relations with the dominant society; (2) comes from below, from the communities themselves and not from public policy, because it is not in the interest of the state for it to work; and (3) promotes processes of decolonization of the subjects, to help them face society on equal terms.

The above are linked to findings in research that refer to cultural variations in the development of socioemotional skills, in spaces of social and cultural diversity in the Mapuche context. The study by Riquelme et al. [52] shows that the school does not consider the relevance of social and emotional elements and the variation of these among students from different cultures. This is because the school system does not contemplate an understanding of the socioemotional processes that are at the base of cultural interactions in spaces of diversity, which are part of the behavior of the subjects. Research conducted by Riquelme et al. [53] clearly identified the beliefs that form the basis of Mapuche knowledge regarding knowing how to feel. From a sociohistorical perspective, these elements have been ignored by the Chilean formal education system in order to maintain a common emotional ideal for all students [53,54]. The results place value on the contribution of the Mapuche family in the identification of the elements that make up their belief system.

Due to this, the issue addressed by the study is the lack of knowledge regarding cultural variations in the perception of school violence in spaces of social and cultural diversity in the Mapuche context. In addition to the above, this research is necessary given the aforementioned negative effects that violence has on the physical and mental health of children and young people, and on the learning outcomes of students. The general objective of the study is to explore the potential variations in the perception of school violence, depending on the ancestry of the students (Mapuche, non-Mapuche), to contribute to the understanding of the phenomenon of peer abuse in a context of social and cultural diversity. This (1) describes the characteristics of school violence perceived by students in relation to differences based on ancestry; and (2) characterizes the variations in perception, according to ancestry.

This study proposes the following hypotheses for all the dimensions of violence and school coexistence:

**H1.** 
*There are statistically significant differences between Mapuche and non-Mapuche students regarding the perception of school violence.*


**H2.** 
*There are statistically significant differences between Mapuche and non-Mapuche students regarding the perception of management school coexistence.*


**H3.** 
*Mapuche students perceive more school violence than non-Mapuche students.*


## 3. Materials and Methods

This is a cross-sectional quantitative study with a descriptive–comparative design [55,56].

### 3.1. Participants

A total of 1404 students from the Chilean school system participated. These students attend six different schools, two schools from each of the three different administrative types of school systems (municipal or local public school, subsidized private school and private school). At the time the instrument was applied, the subjects were between 10 and 13 years old (M = 11.4; SD = 1.1) and were in fifth to eighth grade in urban elementary schools in the city of Temuco. Municipal schools are free, private schools are paid, and subsidized private schools fall under two types of payment models: (1) the first type receives a full state subsidy and are free for families; and (2) the second type receives a state subsidy as well as a co-payment from the parents. One of each type of subsidized private school participated in the study. The cost for families in the co-payment school model is approximately one third of the value of the private schools included in the research. Of all participants, 9.4% were Mapuche and 61% non-Mapuche, based on self-identification according to ancestry. It is important to highlight that the group labeled as ‘unknown’ (29%) corresponds to students who stated that they did not know whether or not they belong to the Mapuche ethnic group. Table 1 provides details of the characteristics of the sample group. Table 2 provides the distribution of Mapuche students in relation to school type.

### 3.2. Instrument

The school coexistence for non-violence (CENVI, Convivencia escolar para la no violencia) questionnaire from Muñoz et al. [4] was applied. This instrument explores the perception of students in relation to school violence and coexistence management. Table 3 presents the structure.

The study by Muñoz et al. [4] reports a good data fit to the model, justifying evidence in favor of structural validity (X^2^ = 7993.75; DF = 2993; CFI = 0.912; TLI = 0.91; RMSEA = 0.033). It also presents good reliability indexes as internal consistency, measured by Cronbach’s alpha. Table 4 presents these indexes.

The CENVI questionnaire is a Likert-type self-response scale, where the respondents must indicate the frequency of occurrence of the events stated in each item. The items present four response options, where: 1 = Never; 2 = Rarely; 3 = Frequently and 4 = Always. The model proposed by Muñoz et al. (2017) is made up of two s-order factors with three and five first-order factors, respectively (hereinafter, dimensions).

For factor 1, ‘types of school violence’, the items were written in negative direction, where the higher the score, the greater the expression of violence. It explores the perception of students in relation to the forms of manifestation of violence in school contexts. It has 49 items organized into five dimensions.

For factor 2, ‘coexistence management’, the items were written in positive direction, where the higher the score, the greater the favorable actions and practices to promote nonviolence. It explores the perception of the students, tied to the central aspects of school management defined by public policy on matters of coexistence, such as comprehensive education, assurance of abuse-free environments, participation and democratic life. It includes 30 items organized into three dimensions.

For conceptual reasons and to economize words, the dimension violence through exclusion is changed to social exclusion, violence through technology is called digital violence and teacher violence towards students is changed to teacher violence. Likewise, based on the Tapia et al. [57] proposal, the dimension management for non-violence is called assurance.

Table 5 and Table 6 describe the situations related to each dimension of factors 1 and 2, respectively. These include the abbreviations that are used hereinafter.

## 4. Procedure

The research project underlying this study was reviewed and authorized by the Research Ethics Committee of the Universidad Católica de Temuco (Chile). The participating schools provided time during school hours to apply the instrument as an activity linked to a reflection on violence and school coexistence. Parents received an informed consent and confidentiality notice and authorized the participation of their children in the research. The students received an informed assent and participated voluntarily. In both cases, they were informed about the characteristics of the research, the instrument and the approximate time it would take to respond. The researchers emphasized the voluntary nature of participation and the anonymity of the students and the school regarding the handling of information and the publication of results. All participants answered the questionnaire online, from computers or tablets at the respective school, in a period not exceeding 15 min. Each student participated simultaneously with their respective grade, accompanied and advised by a teacher and/or school coexistence officer at the school.

### Plan for Analysis

Given the context where the instrument was applied, the researchers considered it pertinent to review the questionnaire model and the content of the items through expert inter-judgment, assessing the possible need to transfer, retain or eliminate items. Expert judgment, according to Robles and Rojas [58], represents a widely used strategy with several advantages. These include ‘(…) determining knowledge about content and difficult, complex and novel or little studied topics’ [59].

The normality of each item was evaluated using the Kolmogorov–Smirnov test to choose the subsequent methods of analysis. The study confirms that the variables do not follow a normal distribution if the statistical significance shows values less than 0.05 [60]. Following the approach of Finney and Distefano [61], researchers carried out CFA using the polychoric correlation matrix and the unweighted least squares mean and variance (ULSMV) estimation method. Goodness-of-fit indicators were calculated, such as the chi-square statistic, the ratio between chi-square and its degrees of freedom, which must be less than 3:1 to indicate a good fit [62,63]. This also included the root mean square error of approximation (RMSEA), which is considered excellent if the value is less than 0.05, and the comparative fit index (CFI), whose optimal values are greater than 0.95, as indicated by Batista-Foguet et al. [64] and Brown [65]. The CFI value can range from 0.9 to 0.95 if the RMSEA value is less than 0.05 [66,67]. Finally, the Tucker–Lewis index (TLI) identifies ideal values greater than 0.95 and acceptable values greater than 0.9 [66,67,68].

The study uses the model proposed by Muñoz et al. [4], which is composed of two s-order factors and eight first-order factors (5 and 3, respectively). The study follows the approach of Lévy and Varela [69] and reviews the correlations between the latent variables in a first-order confirmatory model. In this, a correlation measure less than 0.5 indicates discriminant validity and greater than 0.85 is evidence of convergent validity between factors, which could imply the need to respecify the structure, as long as the result is a significant model. According to Levy et al. [69], when there are high correlations between first-order factors and when the model underlies specific theoretical concepts, it is possible to carry out second-order CFAs. Thus, the first-order latent variables make up one or more second-order factors, highlighting that ‘(…) in the second-order CFA, the assumptions do not change with respect to the first-level CFA measurement model’ [70].

In addition to the above, a respecified model is only adopted if improvements are obtained in all the indicators named for a CFA, which would prove a parsimonious fit [71]. This would specifically be for CFI increases greater than 0.01 and RMSEA decreases greater than 0.015 [72].

The convergent validity of the instrument was evaluated as proposed by Hair et al. [71], which includes the following three elements for each dimension: (1) standardized loads whose values must be greater than 0.5 and their level of statistical significance with a *p*-value less than 0.05; (2) composite reliability where values must be greater than 0.7; and (3) average variance extracted (AVE), requiring values greater than 0.5.

Composite reliability was used to estimate reliability, using the McDonald [73] omega coefficient, identifying values greater than 0.65 as admissible, values between 0.7 and 0.9 as acceptable, and values greater than or equal to 0.9 as excellent [74].

The structure of the scale was first studied to compare the perception of violence between groups, using CFA for each group of defined categories. Measurement invariance analyses were then carried out based on categories (ancestry, sex, school type, payment, grade and age), using configural, metric and scalar models. The configural model was defined as achieved if the goodness-of-fit indicators were met according to the criteria already mentioned for a CFA. Both metric and scalar invariance were considered to be achieved if one or more of the variation criteria established by Cheung and Rensvold [75] and Chen [72] were met. This would mean a variation of RMSEA less than 0.015, CFI less than 0.01 and TLI less than 0.01. Regarding the above, in relation to the relatively small number of Mapuche students who participated in the study, Chen [72] ‘suggested that change in CFI ≤ |−0.005| and change in RMSEA ≤ 0.010 for unequal sample size with each group smaller than 300’.

Once all of the above had met, it was possible to make unbiased comparisons between the results of the CENVI questionnaire with respect to the groups. Given the objective of this study, the differences in the perception of violence according to ancestry were explored through hypothesis contrast tests, selected based on the type of variables, distribution and size of the sample. Since the distribution of the variables did not resemble the normal distribution, the Kruskal–Wallis test was applied to determine the existence of statistically significant differences between three or more groups. The Mann–Whitney U test was then used for the pairwise comparison [56].

All analyses were performed with SPSS version 27 [76], MPlus version 8.8 [77] and Excel version 2206 [78] software.

## 5. Results

The review by expert inter-judgment regarding the content of the indicators led to the decision to eliminate the following items: numbers 1 and 3 of the verbal violence dimension; number 47 of the teacher violence dimension; and numbers 57 and 61 of the education dimension, retaining a total of 74 items (Table 7). The Kolmogorov–Smirnov test for normality rejected the null hypothesis for all items (*p* < 0.001).

The goodness-of-fit indexes considered in the CFA show that the proposed model has a good fit with the data: X^2^ = 7304.699; DF = 2618; RMSEA = 0.036; CFI = 0.914; TLI = 0.911. The dimensions of factor 1 presented correlation measures from 0.6 to 0.9. The dimensions of factor 2 presented correlations from 0.6 to 0.8.

All items presented values higher than 0.5 with statistical significance (*p* < 0.001). The composite reliability for each dimension is ω > 0.89. The AVE shows values greater than 0.5. Therefore, it is possible to affirm convergent validity. Likewise, reliability indicators are excellent, as some dimensions reach a value of ω = 0.9 and the others have higher values (Table 8).

The CFA for the groups of the categories explored shows good indicators, as seen in Table 9.

Table 10 shows invariance analysis by ancestry (the categories are: non-Mapuche; unknown; Mapuche), which shows that the differences in RMSEA, CFI and TLI do not show relevant changes in fit. This affirms that the data meets configural, metric and scalar invariance, which occurs in the same way in the invariance analysis for sex, school type, payment, grade and age range (Table 11).

A comparison of the results of each dimension based on ancestry groups maintains a null hypothesis that the median is the same between the categories. The alternative hypothesis is that there are statistically significant differences between the medians of the ancestry categories. The Kruskal–Wallis test for independent samples found statistically significant differences in the verbal violence, physical violence and teacher violence dimensions (Table 12).

For a deeper look into the differences found, the authors reviewed the dimensions that showed statistically significant differences in the previous test. A pairwise comparison was made through the Mann–Whitney U test, showing that the dimensions of verbal violence and physical violence present statistically significant differences between non-Mapuche and Mapuche people, as well as between non-Mapuche people and unknown. The teacher violence dimension only shows a statistically significant difference between non-Mapuche people and unknown (Table 13).

To identify which categories of ancestry perceive greater violence in the dimensions where statistically significant differences were found, the authors reviewed average ranges of the raw scores. Table 13 shows that Mapuche students perceive more verbal and physical violence than non-Mapuche students. Students who do not know their ancestry perceive more verbal and physical violence than non-Mapuche students. Non-Mapuche students perceive greater teacher violence than students who do not know if they belong to the Mapuche people.

## 6. Discussion and Conclusions

This research provides a valid tool for an unbiased comparison on the perception of violence and management of school coexistence in the groups explored, because it presents empirical evidence that guarantees that the instrument has the same meaning in the groups compared. The above was possible using factorial invariance analysis, a procedure that is not used in several studies that compare variables between two or more groups [79].

The results show that in the Chilean school system there is a lack of knowledge on the perception of school violence in consideration of variations among students who identify with another culture. This is related to the results of the study by Muñoz et al. [7], which showed that indigenous children develop in social and school contexts marked by expressions of structural violence, which, for authors such as Muñoz et al. [40] and Galtung [9], limit the possibilities for the vital development of subjects belonging to subalternized groups. This study explored the differences in the perception of Mapuche and non-Mapuche students in relation to school violence and coexistence management.

To meet the proposed objectives, the study verified that there are statistically significant differences in the perception of verbal and physical violence based on ancestry, with Mapuche students perceiving more violence than non-Mapuche students. Based on the theory of structural violence [9], this finding shows evidence of violence rooted in the Eurocentric colonial and monocultural matrix of the Chilean school system, which, as a tool of power, favors the submission of the Mapuche subject to the hegemonic society [80,81]. Similarly, the lack of recognition of the ‘Mapuche’ in the Chilean school system can be related to the studies carried out by Riquelme et al. [52], Riquelme et al. [53] and Halberstadt et al. [54]. These authors have shown that schooling in the Mapuche context does not consider cultural variations in the emotional development of children, ignores the beliefs of the Mapuche culture in terms of knowing how to feel, and encourages a common emotional ideal for all students.

The finding related to the perception of verbal violence is linked to ridicule, insults and threats, and this violence can be manifested explicitly or implicitly. The verbal violence perceived by Mapuche students may be related to discriminatory practices exercised by their peers or teachers [82]. In keeping with Ortiz-Velosa et al. [82], Mapuche students have historically been the object of stigmatization or ridicule for their school performance, expression of their culture, physical appearance or their last name. Likewise, the findings regarding verbal violence agree with the results of research conducted by Muñoz et al. [7], which revealed that to date there are specific manifestations of verbal violence towards the Mapuche student. These involve not recognizing Mapuche people as belonging to an indigenous group, instead calling them Indians or Black and upholding a recurrent prejudice regarding their cognitive abilities. Saiz et al. [43] found meta-stereotypical labels and attributes that arise from the perception of Mapuche subjects from the city of Temuco (Chile). This study identified the ‘Mapuchito’ label, which contains the attributes of lazy, dirty, drunk, conflictive, poor and stupid. It also identified the ‘Indian’ label, which includes attributes such as incapable, primitive, inferior and ignorant.

The finding regarding the perception of physical violence relates to the use of force against the other, or when damage is caused to the victim’s belongings. In both cases, the objective is domination and humiliation, although on some occasions the aggressors’ objective is to assert themselves in their environment [17]. This finding is linked to research conducted by Romero et al. [83], which points out that physical violence in the school and indigenous context can occur as a response to frustration due to adaptation processes, regulation of emotions and capacities to reach agreements. It is possible to consider that Mapuche students perceive physical violence differently, because they are immersed in a monocultural educational system that constantly exercises structural symbolic violence [82,83]. The study conducted by Muñoz et al. [7] showed a peaceful tendency in this ethnic group in terms of conflict resolution in the school environment, in contrast to non-Mapuche students. This could explain the greater perception of physical violence perceived by Mapuche students, considering the evidence, on the one hand, that in the education of people in the family and community environment in the Mapuche context, there is a particular underlying value of upstanding and respectful behaviors towards other people [84,85]. There are also anti-values and models of the undesired person, linked to violence in community coexistence [49]. In a broader sense, Alonqueo et al. [86] demonstrated that the value of respect for Mapuche girls transcends the human being and encompasses the natural world of non-human living beings.

The results show a significant number of students who do not identify themselves culturally as Mapuche or as non-Mapuche. Although the instrument does not explain why there is no cultural identification, some studies indicate that the school education system has placed tension on the cultural identity of Mapuche children, leading them to assimilate [49,82]. Students who perceive explicit or implicit discrimination, understood as a form of verbal violence, choose to abandon identity aspects that characterize them as Mapuche students [82]. This is the intended effect of the assimilation policies imposed by the Chilean State to form the idea of superiority of the Western European over the indigenous and of the urban over the rural [34,84]. As a result of this, Mapuche students interact in a system that mobilizes their deculturation as indigenous and that promotes their sense of belonging to the state, while abandoning their sense of belonging to their culture. Proof of this can be seen in the findings of Mansilla and Imilan [87], which reveal mechanisms of the Chilean State for the territorial dispossession of the Mapuche people, who are finally pressured into denying their status as indigenous, their last names and the practice of their traditions. However, there is no way of knowing why a student fails to self-identify as belonging or not belonging to the Mapuche people.

The findings made are relevant, as there is no other research available in a Mapuche context specifically regarding the objectives and hypotheses raised in this research. Furthermore, these concur with various international studies regarding school violence, discrimination and health. Peguero and Jiang [88] found high positive correlations between ethnicity and victimization, showing that young people who belong to an ethnic group and interact with those who do not belong suffer more school violence than other students. A study on school dropouts carried out by Peguero [89] concluded that students belonging to ethnic groups that suffer from school violence have a higher risk of abandoning their studies. In this sense, indigenous students with strong cultural roots have a higher risk of dropping out of school, since their language is not used in schools, which results in a negative impact on their academic performance [90]. In the field of health, the evidence shows that people belonging to an ethnic group who perceive discrimination against them have an increased risk of suffering from physical and mental pathologies, although it is important to note that the positive correlation between discrimination and illness is stronger in subjects facing discrimination due to gender, age and sexuality [91]. This leads us to ask—as a minimum—if the perception of discrimination in an indigenous person is greater, depending on their gender, age and sex.

Although international studies have been conducted that address indigenous students’ perceptions of violence—and safety, this study is delimited to the Chilean school system and particularly to the Mapuche context. In other countries, there is evidence from qualitative studies that have been conducted that allow us to sustain that, in spaces of social and cultural diversity, hegemonic relations predominate in a logic of structural violence. Based on this, it is plausible to say that these power imbalances between the dominant and dominated society have represented a reality in school spaces with the presence of indigenous students. A study conducted by Burrage et al. [92] describes the case of Indian residential schools (IRS) in Canada, where indigenous children were removed from their communities and placed in the IRS, where they suffered psychological, physical and sexual abuse. To this day, this has consequences not only for the surviving victims, but also for their families and indigenous communities, because it has prevented cultural transmission and resulted in the loss of their language. A study by Whitehead [38] reviews the school system imposed in the British colonies after World War II, which managed to impose and reinforce the cultural superiority of Great Britain by sending mostly female teachers. This implied the transformation of some colonies into places that are predominantly divided along class, gender, race and nationality lines. Chilisa and Mertens [93] reveal the existence of an epistemic entrenched racism and detrimental treatment, giving rise to discrimination through education, imposing the forms of construction of knowledge and symbols of the dominant culture. This is detrimental to the cultural roots of the involuntarily dominated and minority groups in different countries. This is the case with the Mapuche people, as seen in this study, where the Chilean State fails to consider the components of Mapuche family education [35,84]. Such elements could very well contribute to the peaceful resolution of conflicts in the school setting [7] by taking cultural variations into account in the processes of emotional socialization [52,53,54].

In relation to the proposed hypotheses, Hypothesis 1 is confirmed, due to the evidence of statistically significant differences between Mapuche and non-Mapuche students regarding the perception of school violence. Hypothesis 2 is rejected, given that there are no statistically significant differences between Mapuche and non-Mapuche students regarding the perception of management of school coexistence. Hypothesis 3 is confirmed, as the evidence shows that Mapuche students perceive more school violence than non-Mapuche students. However, regarding Hypothesis 3, differences were found in the dimensions of physical violence and verbal violence.

In light of the findings, educational centers located in areas of social and cultural diversity in the Mapuche context should consider adaptations in the management of school coexistence aimed at reducing and preventing school violence. This is particularly the case for elements that affect verbal and physical violence related to interactions between Mapuche and non-Mapuche students. The actions incorporated can be addressed locally by schools, since current regulations in Chile allow educational administrators to promote decentralization and improve the relevance of public policy through their management [94]. However, changes in the management of coexistence must not omit the knowledge produced in recent years regarding education in a Mapuche context. It is critical that the variations in socioemotional development between students from different cultures are taken into account [52,53,54], and that the integration of the family and the community is established as part of the educational task [48].

The projection of the study lies in the possibility of replicating it in other local and regional school contexts, to explore the variations in perception regarding each dimension of the instrument. Future analyses may consider comparison by age, sex, grade, school type, funding and the socioeconomic status of the students. Likewise, a future study may contemplate methodological complementarity including qualitative designs and instruments to delve deeper into the notion of school violence that underlies students’ perceptions.

Among the study’s limitations, which may also represent an area for improvement in future research, is the lack of knowledge of the characterization of the group that labels its ancestry as ‘unknown’. Because of this, we cannot be certain about the reasons why some students say that they do not know whether they belong to the Mapuche ethnic group. Furthermore, given the sample size and the fact that this sample is taken from a single city, the findings cannot be generalized for the entire school population in the country. However, the findings can be considered valuable for the context of the city of Temuco and potentially for the region of La Araucanía, since it is the region with the highest proportion of Mapuche people in the country with respect to its total population.

## Figures and Tables

**Figure 1 ijerph-20-00024-f001:**
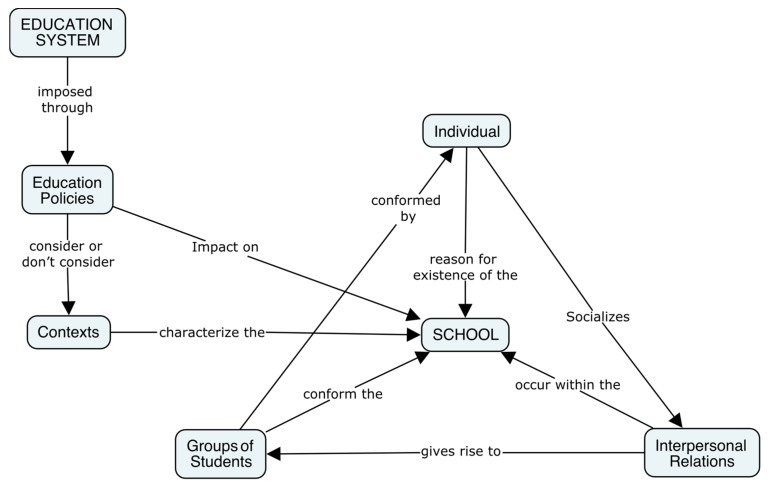
Levels of school violence. Source: prepared by the authors.

**Figure 2 ijerph-20-00024-f002:**
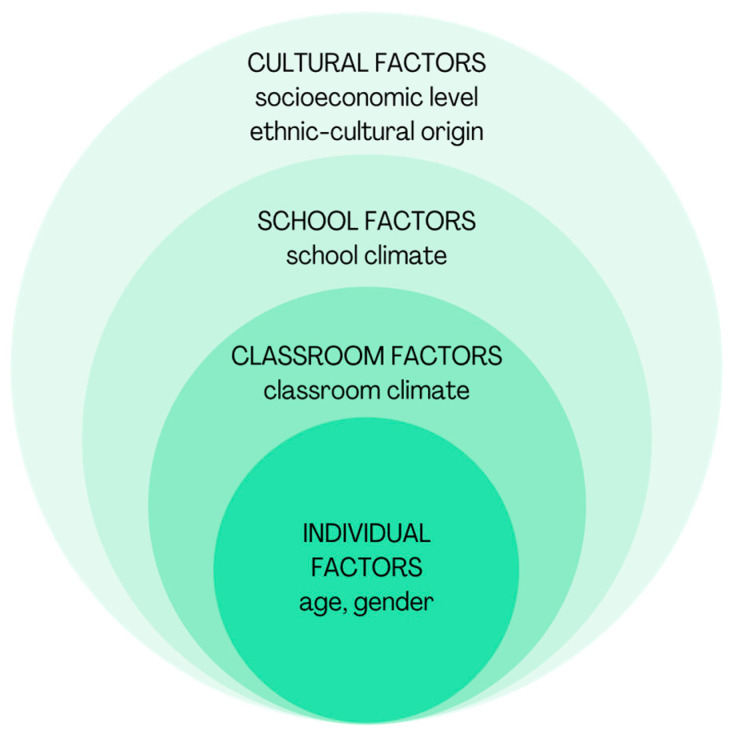
Levels of school violence. Source: López et al. [28].

**Table 1 ijerph-20-00024-t001:** Characteristics of the participants. Source: prepared by the authors.

		Frequency	Percentage
Sex	Male	725	51.6
	Female	679	48.4
Age Range	11 years or younger	713	50.8
	12 years or older	691	49.2
Ancestry	Non-Mapuche	865	61.6
	Unknown	407	29.0
	Mapuche	132	9.4
Grade	5th grade	375	26.7
	6th grade	338	24.1
	7th grade	377	26.9
	8th grade	314	22.4
School Type	Municipal or local public	430	30.6
	Subsidized private	561	40.0
	Private	413	29.4
Payment	Paid	688	49.0
	Free	716	51.0

N = 1404.

**Table 2 ijerph-20-00024-t002:** Distribution of ancestry based on school type. Source: prepared by the authors.

	Non-Mapuche	Unknown	Mapuche	Total	Percentage
Municipal or Local Public	228	140	62	430	14.4%
Subsidized Private	298	203	60	561	10.7%
Private	339	64	10	413	2.4%

**Table 3 ijerph-20-00024-t003:** Structure of CENVI questionnaire. Source: Muñoz et al. [4].

Factors	Dimensions		Abbreviation	Items by Dimension	Item No.
Factor 1 Types of violence	1	Verbal violence	VV	10	1–10
	2	Physical violence	PV	11	11–21
	3	Violence through social exclusion	VS	10	22–31
	4	Violence through technology	VT	9	32–40
	5	Teacher violence towards students	TV	9	41–49
Factor 2 Coexistence management	1	Education	EV	12	50–61
	2	Management for non-violence	MV	10	62–71
	3	Participation	PA	8	72–79

**Table 4 ijerph-20-00024-t004:** Structure of CENVI questionnaire. Source: Muñoz et al. [4].

Factors	Dimensions	Cronbach’s Alpha	Median %	Variance %
Factor 1 Types of violence	VV	0.810	13.39	35.5%
	PV	0.886	10.21	43.3%
	VS	0.902	10.93	46.5%
	VT	0.886	10.21	43.0%
	TV	0.911	5.76	39.2%
Factor 2 Coexistence management	EV	0.853	7.49	50.6%
	MV	0.896	18.14	54.1%
	PA	0.890	18.20	54.9%

**Table 5 ijerph-20-00024-t005:** Actions related to the dimensions of factor 1. Source: Muñoz et al. [4].

Dimensions	Abbreviation	
Verbal violence	VB	Aggressions through words such as insults, threats, offensive nicknames.
Physical violence	PV	Pushing and shoving, hair pulling, pinching, punching and kicking or hitting with objects. Indirect physical violence occurs when it is perpetrated on the belongings or work materials of the victim.
Social exclusion	SE	Acts of discrimination or rejection based on academic performance, nationality, cultural or ethnic differences, physical characteristics or personal appearance.
Digital violence	DV	Aggressions through mobile phones or other communication devices via the internet, through photos, videos or text messages.
Teacher violence	TV	Teacher aggressions towards the student, whether verbal, physical or discriminatory.

**Table 6 ijerph-20-00024-t006:** Actions related to the dimensions of factor 2. Source: Muñoz et al. [4].

Dimensions	Abbreviation	
Education	ED	Practices for reflection and education based on dialogue, respect and legitimate acceptance of the other, in order to reduce the risk of situations of violence.
Assurance	AS	Construction and compliance with rules of coexistence for the prevention, control and punishment of violence.
Participation	PA	Actions aimed at the integration of members of the educational community, to contribute to the construction of safe spaces that are free from abuse.

**Table 7 ijerph-20-00024-t007:** Structure of CENVI questionnaire, 74 items. Source: prepared by the authors.

Second-Order Factors	First-Order Factors (Dimensions)		Abbreviation	Items by Dimension	Item No.
Factor 1 Types of violence	1	Verbal violence	VV	8	1–8
	2	Physical violence	PV	11	9–19
	3	Social exclusion	SE	10	20–29
	4	Digital violence	DV	9	30–38
	5	Teacher violence	TV	8	39–46
Factor 2 Coexistence management	1	Education	ED	10	47–56
	2	Assurance	AS	10	57–66
	3	Participation	PA	8	67–74

**Table 8 ijerph-20-00024-t008:** Convergent validity indicators. Source: prepared by the authors.

Factors	Dimensions	Range of Correlations Minimum; Maximum	Compound Reliability	AVE
Coexistence	Verbal violence	0.578; 0.829	0.894	0.516
	Physical violence	0.615; 0.873	0.926	0.536
	Social exclusion	0.595; 0.831	0.912	0.511
	Digital violence	0.728; 0.848	0.943	0.650
	Teacher violence	0.570; 0.838	0.895	0.521
Violence	Education	0.582; 0.804	0.912	0.510
	Assurance	0.643; 0.805	0.917	0.528
	Participation	0.623; 0.782	0.892	0.510

Note. All the items showed a statistical significance of *p* = 0.000.

**Table 9 ijerph-20-00024-t009:** CFA groups of the categories reviewed. Source: prepared by the authors.

Category	Group	N	X^2^	DF	RMSEA	CFI	TLI
Ancestry	Non-Mapuche	865	5383.348	2599	0.035	0.913	0.909
	Unknown	407	3415.274	2599	0.028	0.944	0.942
	Mapuche	132	2883.243	2599	0.029	0.942	0.940
Sex	Male	725	4817.809	2599	0.034	0.921	0.917
	Female	679	4415.148	2599	0.032	0.929	0.926
School Type	Municipal or local public	430	3655.251	2599	0.031	0.931	0.928
	Subsidized private	561	3774.791	2599	0.028	0.941	0.938
	Private	413	3335.064	2599	0.026	0.927	0.924
Payment	Free	688	4231.881	2599	0.030	0.922	0.919
	Paid	716	4154.266	2599	0.029	0.946	0.943
Grade	5th grade	375	3309.719	2599	0.027	0.940	0.937
	6th grade	338	3370.381	2599	0.030	0.932	0.929
	7th grade	377	3705.927	2599	0.034	0.917	0.914
	8th grade	314	3777.217	2599	0.038	0.913	0.910
Age	11 years or younger	713	4196.680	2599	0.029	0.935	0.932
	12 years or older	691	4964.290	2599	0.036	0.916	0.912

**Table 10 ijerph-20-00024-t010:** Measurement invariance test based on ancestry. Source: prepared by the authors.

	X^2^	DF	*p*	RMSEA	CFI	TLI	ΔX^2^	ΔDF	ΔRMSEA	ΔCFI	ΔTLI
Configural	9039.684	7797	0.000	0.018	0.942	0.940	---	---	---	---	---
Metric	9148.334	7929	0.000	0.018	0.943	0.942	108.650	132	0.000	0.001	0.002
Scalar	9374.489	8209	0.000	0.017	0.946	0.946	226.155	280	−0.001	0.003	0.004

**Table 11 ijerph-20-00024-t011:** Measurement invariance test, other categories. Source: Prepared by the authors.

		X^2^	DF	*p*	RMSEA	CFI	TLI	ΔX^2^	ΔDF	ΔRMSEA	ΔCFI	ΔTLI
SEX	Configural	9237.327	5198	0.000	0.033	0.924	0.921	---	---	---	---	---
	Metric	9342.896	5264	0.000	0.033	0.923	0.921	105.569	66	0.000	−0.001	0.000
	Scalar	9445.358	5404	0.000	0.033	0.924	0.924	102.462	140	0.000	0.001	0.003
SCH	Configural	10,510.513	7797	0.000	0.027	0.937	0.935	---	---	---	---	---
	Metric	10,866.019	7929	0.000	0.028	0.932	0.930	355.506	132	0.001	−0.005	−0.005
	Scalar	11,474.656	8209	0.000	0.029	0.924	0.925	608.637	280	0.001	−0.008	−0.005
PAY	Configural	8358.142	5198	0.000	0.029	0.936	0.933	---	---	---	---	---
	Metric	8591.323	5264	0.000	0.030	0.933	0.931	233.181	66	0.001	−0.003	−0.002
	Scalar	8968.137	5404	0.000	0.031	0.928	0.928	376.814	140	0.001	−0.005	−0.003
GRA	Configural	14,078.571	10,396	0.000	0.032	0.926	0.923	---	---	---	---	---
	Metric	14,368.633	10,594	0.000	0.032	0.924	0.923	290.062	198	0.000	−0.002	0.000
	Scalar	14,815.461	11,014	0.000	0.031	0.924	0.925	446.828	420	−0.001	0.000	0.002
AGE	Configural	9086.815	5198	0.000	0.033	0.926	0.923	---	---	---	---	---
	Metric	9220.049	5264	0.000	0.033	0.925	0.923	133.234	66	0.000	−0.001	0.000
	Scalar	9391.125	5404	0.000	0.032	0.924	0.924	171.076	140	−0.001	−0.001	0.001

Note. SEX = sex; SCH = school; PAY = payment; GRA = grade; AGE = age.

**Table 12 ijerph-20-00024-t012:** Kruskal–Wallis test, ancestry category. Source: prepared by the authors.

Dimensions	Test Statistic	Statistical Significance
Verbal violence	20.099	<0.001 *
Physical violence	11.992	0.002 *
Social exclusion	4.016	0.134
Digital violence	0.102	0.950
Teacher violence	14.017	0.001 *
Education	0.141	0.932
Assurance	1.180	0.554
Participation	1.182	0.554

* Statistical significance at the 0.01 level.

**Table 13 ijerph-20-00024-t013:** Pairwise comparison with Mann–Whitney U test. Source: prepared by the authors.

	Group 1	Range Average	Group 2	Range Average	*p*
Verbal Violence	non-Mapuche	487.95	Mapuche	571.38	0.002 *
	non-Mapuche	609.85	unknown	693.14	<0.001 *
	Mapuche	280.32	unknown	266.65	0.380
Physical Violence	non-Mapuche	490.29	Mapuche	556.06	0.014 **
	non-Mapuche	616.25	unknown	679.55	0.004 *
	Mapuche	280.32	unknown	266.65	0.380
Teacher Violence	non-Mapuche	501.54	Mapuche	482.34	0.437
	non-Mapuche	663.63	unknown	578.83	<0.001 *
	Mapuche	287.20	unknown	264.42	0.143

* Statistical significance at the 0.01 level; ** Statistical significance at the 0.05 level.

## Data Availability

Data available at the Appendix A.

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
