# Peer review of "Perception of School Violence: Indicators of Normalization in Mapuche and Non-Mapuche Students"

_ijerph, 2022, doi:10.3390/ijerph20010024_

Round 1

Reviewer 1 Report

Thank you for presenting an interesting paper on an important topic. The focus on the Chilean education system in one city is reasonable, though it does create some limitations for generalisation of the findings.

At line 85, you talk about three seemingly discrete players, aggressor, victim and system. I'd suggest that the victim can also be the perpetrator/aggressor. 

At line 95 I think you need a reference for your assertion.

Line 104, you talk about 'spheres'. What are they?

Line 175, you talk about the harmful effects on subjects of violence. While this is true, I suggest that the harmful effects of entrenched violence extend intergenerationally and so has long term impacts tht extend to future generations.

Discussions and conclusion. I think you should also include a response to your hypotheses in this section, so that it is clear whether or not they are confirmed.

Line 548, you say there are no other studies found in other cultures. There are many studies that have explored this issue from different angles, including the comparative angle you have taken. You might consider changing your comment here to suggest that 'while other international studies exist in relation to indigenous students perceptions of violence (and safety) this study is delimited to the Chilean context.'

Limitations: I think the other key limitation is the relatively small number of Mapuche students in the study, which surely must make general assertions difficult to make.

You havent made a comment about what your findings might mean for violence prevention or anti-violence interventions. Can you make some comment about this.

Author Response

Dear Reviewer.
Thanks for your comments. I will answer you point by point.

(1) Thank you for presenting an interesting paper on an important topic. The focus on the Chilean education system in one city is reasonable, though it does create some limitations for generalisation of the findings.

R: We have included the limitation that you indicate. However, it was important to add that of all the regions in Chile, La Araucanía has the highest proportion of Mapuche people in relation to the total population.

(2) At line 85, you talk about three seemingly discrete players, aggressor, victim and system. I'd suggest that the victim can also be the perpetrator/aggressor. 

R: We have considered this clarification that the victim can also be an aggressor in the subsequent lines.

(3) At line 95 I think you need a reference for your assertion.

R: These are now included. We thought that it wasn’t necessary since these lines provide a succinct summary of the previous ideas.

(4) Line 104, you talk about 'spheres'. What are they?

I am not sure if ‘spheres’ was the most appropriate translation, and therefore we have changed this to ‘spaces’. In the context of the paragraph, I indicate that the ‘environments’ or ‘spaces’ will be differentiated or autonomous, based on the nature of the categories and the relationships of these based on their levels, which is also addressed in the next paragraph.

(5) Line 175, you talk about the harmful effects on subjects of violence. While this is true, I suggest that the harmful effects of entrenched violence extend intergenerationally and so has long term impacts that extend to future generations.

R: This is a very interesting concept that I hadn’t looked into and it has also been incorporated based on your comment.

(6) Discussions and conclusion. I think you should also include a response to your hypotheses in this section, so that it is clear whether or not they are confirmed.

R: This is a very important point, which I had omitted, on the understanding that this was explained without alluding to the hypotheses. In relation to this, we found it necessary to separate the hypotheses on school violence and management of school coexistence. These are now presented as 3 hypotheses and are addressed in the discussion and conclusions section.

(7) Line 548, you say there are no other studies found in other cultures. There are many studies that have explored this issue from different angles, including the comparative angle you have taken. You might consider changing your comment here to suggest that 'while other international studies exist in relation to indigenous students perceptions of violence (and safety) this study is delimited to the Chilean context.'

R: I really appreciate this comment because it was necessary to make this clarification, because this idea was too general. I completely adhere to the suggestion made and have made the appropriate changes to the text.

(8) Limitations: I think the other key limitation is the relatively small number of Mapuche students in the study, which surely must make general assertions difficult to make.

R: Although this can be presented as a limitation, I have added the total sample size as a limitation. Regarding the small number of Mapuche students, I opted to add the following to the plan for analysis: “Chen (2007) suggested that change in CFI <= |-0.005| and change in RMSEA <= 0.010 for unequal sample size with each group smaller than 300.” The results then show differences in CFI and RMSEA that are well below Dr. Chen’s suggestion.

(9) You haven’t made a comment about what your findings might mean for violence prevention or anti-violence interventions. Can you make some comment about this.

R: I have incorporated comments on the importance of management of school coexistence and the necessary consideration of the findings of educational research in a Mapuche context, regarding variations in emotional development between students from different cultures and the incorporation of the family and community.

Reviewer 2 Report

The need for the study should be added and enriched. There is a need to write research questions and findigns should be presented based on research questions. Argumentation in discussion should be added. 

Author Response

Dear Reviewer.
Thanks for your comments. I will answer you point by point.

(1) The need for the study should be added and enriched. There is a need to write research questions and findings should be presented based on research questions. Argumentation in discussion should be added. 

R: A more explicit argumentation has been incorporated regarding the need for the study. Regarding the research questions you mention, they are not addressed as such, because we have defined a research problem and presented hypotheses. We feel that we still address your idea because, based on the comments of the peer reviewer, we have adapted the hypotheses and added responses to these in the discussion and conclusion section.